

# Late Holocene glacier variations in the central Tibetan Plateau indicated by the $\delta^{18}O$ of ice core enclosed gaseous oxygen

Jiule Li[1,2*], Baiqing Xu[1,2,3], Ninglian Wang[3,4], Ping Yao[1], Xiangke Xu[1,2,3]

[1]Institute of Tibetan Plateau Research, Chinese Academy of Sciences, Beijing 100101, China.
[2]Key Laboratory of Tibetan Environment Changes and Land Surface Processes, Chinese Academy of Sciences, Beijing 100101, China.
[3]CAS Center for Excellence in Tibetan Plateau Earth Sciences, Chinese Academy of Sciences, Beijing 100101, China.
[4]College of Urban and Environmental Sciences, Northwest University, Xi'an, 710069, China.

*Correspondence to*: Jiule Li (jlli@itpcas.ac.cn)

**Abstract.** The $\delta^{18}O$ of gaseous oxygen in ice core air bubbles ($\delta^{18}O_{bub}$) has been widely used for reconstruction of climate changes in polar glaciers. Yet, less is known about its climatic implication in alpine glaciers as the lack of continuous record. Here, we present a long-term $\delta^{18}O_{bub}$ record from the Tanggula glacier in the central Tibetan Plateau (TP). It shows that there is a good correlation between the variation of the $\delta^{18}O_{bub}$ in this alpine ice core and the accumulation or melting of the glacier. The more developed the firn layer on glacier surface, the more positive the $\delta^{18}O_{bub}$ is. The more intense the glacier

melting, the more negative the $\delta^{18}O_{bub}$ is. Combined with the chronology of the ice core air bubbles, we reconstructed the glacier variations since the late Holocene in the central TP. It showed that there were four accumulation and three deficit periods of glaciers in this region. The strongest glacier accumulation period was from 1610-300 B.C., which corresponding to the Neoglaciation. The most significant melting period was the last 100 years, which corresponding to the recent global warming. During the Little Ice Age, glacier accumulation in the central TP was not significant, and even short deficit events

occurred. Comparisons of the late Holocene glacier variation in the central TP with hemispheric glacier and climate variations showed that it was closely related to the North Atlantic Oscillation.

## 1 Introduction

Ice core air bubble is natural atmosphere enclosed in the glacial ice during the transition of firn into ice. It provides the most direct way to study the ancient atmosphere environment, and had been widely used for reconstructions of regional or global

climate and environment changes (e.g., Bender, 2002; Capron et al., 2010; Bazin et al., 2016). For instance, studies about the ice core air bubbles revealed that the global climate change were closely related to the variation of greenhouse gases in atmosphere since the last deglaciation (Loulergue et al., 2008; Shakun et al., 2012; Chappellaz et al., 2013; Parrenin et al., 2013; Landais, 2016). The air contents in ice cores from polar and alpine glaciers were dominated by the solar radiation intensity and has largely been used for the establishment of ice core chronologies based on its link with the integrated local

insolation (Raynaud et al., 2007; Parrenin et al., 2007; Li et al., 2011; Eicher et al., 2016). Variations of the elemental ratios





in the ice core gases, such as the $O_2/N_2$ and $Ar/O_2$, could provide valuable information about processes occurring during the bubble close-off process in the firn (e.g., Huber & Leuenberger, 2004; Mitchell et al., 2015 ). Furthermore, variation of the $O_2/N_2$ from Antarctic ice core was proved to be closely related to local summertime insulation and could be used to date old ice (Bender, 2002; Bazin et al., 2016).

The stable isotope ratio ($\delta^{18}O$) of gaseous oxygen in ice core air bubble ($\delta^{18}O_{bub}$) is a complex signal. It could be affected by several factors, such as the global Dole effect (Dole, 1935; Lane and Dole, 1956), the gravitational and thermal fractionation (Huber et al., 2006; Severinghaus et al., 1998) and the isotope exchange reactions during the air enclosing and storing processes in the firn column (e.g., Benson and Krause, 1984; Craig et al., 1988; Luz and Eugeni, 2011). Among them, the global Dole effect are mainly influenced by the variations of the global biosphere productivity and the land surface

temperature and humidity conditions (Bender et al., 1994; Leuenberger, 1997; Malaize et al., 1999; Severinghaus et al., 2009), the total solar irradiation (Bender, 2002; Hoffmann et al, 2004) and the global ice volume and hydrological circulation (eg., Sowers et al., 1989, 1991; Jouzel, 2013). The gravitational and thermal fractionation and the isotope exchange reactions are mainly related to regional climate conditions during the enclosing of gases into ice (McConnell et al., 1998; Huber et al., 2006).

Studies about the $\delta^{18}O_{bub}$ were mostly carried out in polar glaciers. It had been found that the $\delta^{18}O_{bub}$ in polar glaciers varied coherently with the sea level caused by changes in global ice volume (e.g., Shackleton, 2000; Landais et al., 2010; Jouzel, 2013). The resemblance between those variations and the precession signal or mid-June 65 °N insolation had also been highlighted (Petit et al., 1999; Dreyfus et al., 2007; Landais et al., 2010) leading to the use of $\delta^{18}O_{bub}$ to synchronize ice cores from Greenland and Antarctica (e.g., Bender et al., 1994; Extier et al., 2018). On the other hand, because the ice formation

process in the alpine glaciers are more sensitive to regional climate conditions and its variation could led to changes of air component in the ice core, the climatic implication of the $\delta^{18}O_{bub}$ may be different in the alpine glaciers as compared to that in the polar glaciers (Luz and Eugeni, 2011). However, few detailed research has been done on this assumption. A better insight into these processes is needed.

As the third pole, the Tibetan Plateau (TP) preserves the largest alpine glaciers in the middle and low latitudes (Yao, 2012,

2019) and is an ideal site for ice core research (e.g., Thompson et al., 1997; 2006; Yao et al, 1992; 1997). Until now, studies about the ice core air bubble has been carried in many glaciers, such as the Dasuopu glacier (e.g., Xu and Yao, 2001; Li et al., 2011), the Rongbuk glacier (Hou et al., 2007; 2013), etc.. One known work about the $\delta^{18}O_{bub}$ were mainly concentrated on its application in the deep ice dating by comparing with the polar ice cores (Hou et al, 2004). So far, none has been carried out on the continuous record of $\delta^{18}O_{bub}$ in the ice core. The climatic implication of the long-term variation of $\delta^{18}O_{bub}$ in the

TP is still unknown. Therefore, in this study, we used a 190.3 m ice core drilled from the Tanggula glacier in the central TP to investigate the climatic implication of the $\delta^{18}O_{bub}$ and the climate information it contained. This may also provide an available method for the reconstruction of climate change and glacier variations by ice cores in the TP.



## 2 Materials and Methods

### 2.1 The Tanggula ice core and the regional geography

The 190.3 m long ice core used in this study was drilled from the Tanggula glacier in the central TP at an altitude of 5645 m (33°06'36.6" N, 92°04'24.4" E, Fig. 1a and b) in 2004 A.D.. This glacier is about 7 km long, with an area of ~19.35 km². The accumulation area of the glacier ranges from 5600 to 5720 m a.s.l.. The bottom end of the glacier is at 5245 m a.s.l., and the snow line is at ~5600 m a.s.l. (Huang et al., 2013). The Tanggula glacier is located at the transitional zone between the semi-arid continental climate and the warm-humid oceanic climate in the TP. It is also the zone where the Westerly, the Indian

monsoon, and the East Asian monsoon interact with each other in the TP (Gao, 1962) (Fig. 1a). Previous studies show that climate and environment changes in the mid-latitudes of the northern hemisphere (NH) could be sensitively reflected in this region (e.g., Thompson et al., 2006; Zheng et al., 2010; Zhu et al., 2015).

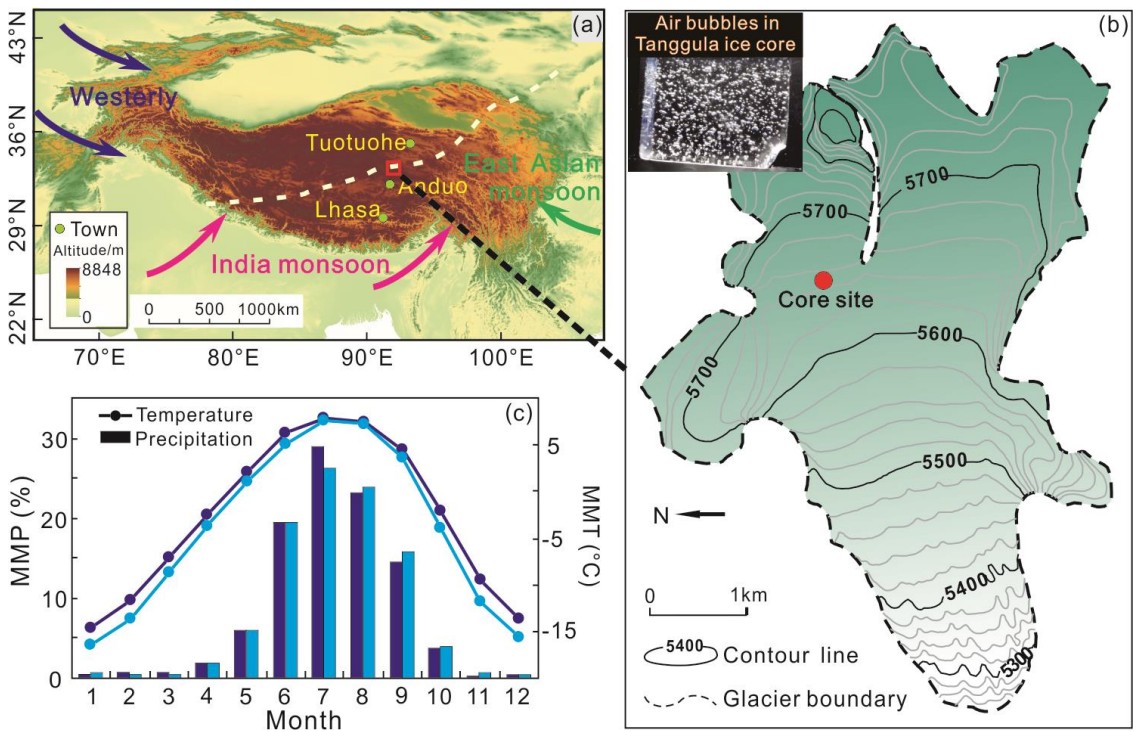

**Figure 1: Geography of the Tanggula glacier. (a) Location of the Tanggula glacier in the TP. The white dashed line**
**indicates the boundary between the Westerly, Indian monsoon and East Asian monsoon. (b) The ice core drilling site and topography of the Tanggula glacier. (c) Temperature and precipitation data from adjacent Anduo and Tuotuohe meteorological stations. MMT = Monthly Mean Temperature. MMP = Monthly Mean Precipitation.**

According to the adjacent Anduo and Tuotuohe meteorological stations, The annual mean precipitation in this regional is about 450 mm. Precipitation mainly occurs from May to October (summer half year) and is quite rare in winter (Fig. 1c).

The regional annual mean temperature is about -6 ℃, with the warmest temperatures in July and August. Recent meteorological observations at the Tanggula glacier showed that the average annual surface temperature around the ice core



drilling site was -9.8 ℃ and the annual temperature difference is 20.9 ℃. The coldest month is January, with a mean monthly temperature (MMT) of -20.6 ℃. The warmest month is August, with an MMT of ~6.1 ℃. The average temperature from June to September is above 0 ℃. Previous studies had also shown that the sand and dust events occur frequently in this

region and were most prevalent from November to May (Zheng et al., 2010).

## 2.2 Experiment and analysis

The manual description and instrumental analysis of the physical properties (i.e., density, snow thickness, air content and contamination layer) of the Tanggula ice core were completed at the State Key Laboratory of Cryosphere Science of China (SKLCS) to help understanding the ice formation process at the core drilling site. First, the ice core was manually observed

in the cryogenic (-20 ℃) cold room. At the same time, digital imagery of each core sample were taken and artificially observed on the computer to estimate the general characteristics of the air content in the Tanggula ice core. During the observation procedure, the ice samples were defined into four grades, which is complete bubble free ice, ice with few bubbles, bubble ice (seen in Fig. 1). The value assigned to each grade was 0, 1, and 2, respectively. Then we calculated the assigned value of each ice core at an interval of 50 cm and converted it into the percentage of air content in ice. The

percentage scale ranged from 0 to 100%, with ice containing a large number of air bubble defined as 100% and pure ice containing no air bubble defined as 0. By this, the general variation characteristic of air content in the 190.3 m Tanggula ice core was determined.

After the analysis of physical properties, the Tanggula ice core was divided into four parts lengthwise. One part was cut by interval of 2 cm. Each sample was cleaned by removing its surface following the cleansing process given by Hou et al. (2003)

and then melted for further detections. The radionuclide β activation in the ice core was detected using a low-concentration α-β activation counting instrument at the SKLCS for absolute age control of the ice. The concentration of the insoluble particles in the ice core was detected using a Multisizer 3 Coulter Counter (Beckman Coulter, Brea, CA, USA). Before the detection, all samples were diluted 5 to10 times with the ISOTON II electrolyte (Beckman Coulter) for the convenience of instrument detection. Soluble inorganic ion concentrations in the upper 27.13 m of the ice core were detected using an ICS

series ion chromatograph (Dian Company, Sunnyvale, CA, USA) at the Key Laboratory of Tibetan Environment Changes and Land Surface Processes (TEL), Chinese Academy of Sciences. Detection accuracy of the conventional anion ($SO_4^{2-}$, $NO_3^-$) and cation ($Na^+$, $Ca^{2+}$, $K^+$) concentrations was <1 ng $g^{-1}$ for both. The stable oxygen ratio ($\delta^{18}O$) of the ice core was also detected at the TEL using an L2130-i Isotope and Gas Concentration Analyzer (Picarro, Santa Clara, CA, USA), with the detection accuracy better than 0.05 ‰.

Air bubbles in the Tanggula ice core were extracted and analyzed at the TEL. Firstly, the air bubbles were extracted via the manual melt-refreeze method (see Xu and Yao, 2001 for detailed procedure). In this work, before the release of air enclosed in the ice, high-speed pure helium (flow rate = 150 mL $min^{-1}$) was used to purge the sample bottle and the gas injection loop to remove the original air and make sure that the ice sample bottle and sampling loop were in pure helium surroundings. After released from the ice, the gases were loaded into a MAT 253™ isotope ratio mass spectrometer (IRMS) (Thermo





Scientific, Waltham, MA, USA) using pure helium (flow rate = 4 mL min$^{-1}$) to detect the $\delta^{18}O_{bub}$, and the detection accuracy was about 0.1‰. To improve the detection accuracy, each gas sample was detected five consecutive times, with the average taken as the final detection value. During the experiment, oxygen from the modern atmosphere was used as the reference gas, and its $\delta^{18}O$ value was designated as 0‰ according to the same detection method given by Petrenko et al. (2006). Meanwhile, the $\delta^{18}O$ values of natural atmospheric oxygen ($\delta^{18}O_{atm}$) from different places and altitudes were also detected for further

discussion of the characteristics and variations of the $\delta^{18}O_{bub}$ in the Tanggula ice core.

## 3 Results and discussion

### 3.1 Chronology of the air bubbles in the Tanggula ice core

The physical properties of the Tanggula ice core showed that only the upper 0.85 m of the ice core is firn, with the remaining core being ice. According to the adjacent Anduo and Tuotuohe meteorological stations, the annual maximum temperature in

this region is 3-5 ℃. Sometimes, the summer temperature could reach 7 ℃ and the winter minimum temperature were about -15 ℃. These data indicate that the area around the Tanggula ice core drilling site is the cold percolation-recrystallization zone (Hou and Qin; 2002). In this kind of the ice formation zone, the surface of the glacier could melt during the summer half year. The meltwater could infiltrate in the firn layer causing the cementation of snow grains. When the temperature decreases in autumn and winter, the aqueous firn could freeze and transform quickly into ice. At the same time, the gases in

the firn are quickly enclosed in the ice. As the natural atmosphere on the surface of the glacier could instantly diffuse through the 0.85 m firn, there should be no significant difference between the age of the ice and the age of its enclosed gases in the same layer of the Tanggula ice core. The age of the ice could be used as the age of its enclosed air bubble. Considering the possible variation of accumulation process on the Tanggula glacier, we concluded that the gas-ice age difference should not exceed 50 years when there were thick firn layer developed on the glacier based on the ice core bore hole gas

measurements at the Dasuopu glacier (Xu and Yao, 2001), the gas-ice age difference calculation at the East Rongbuk glacier (Hou et al., 2007) and the regional present precipitation and temperature.



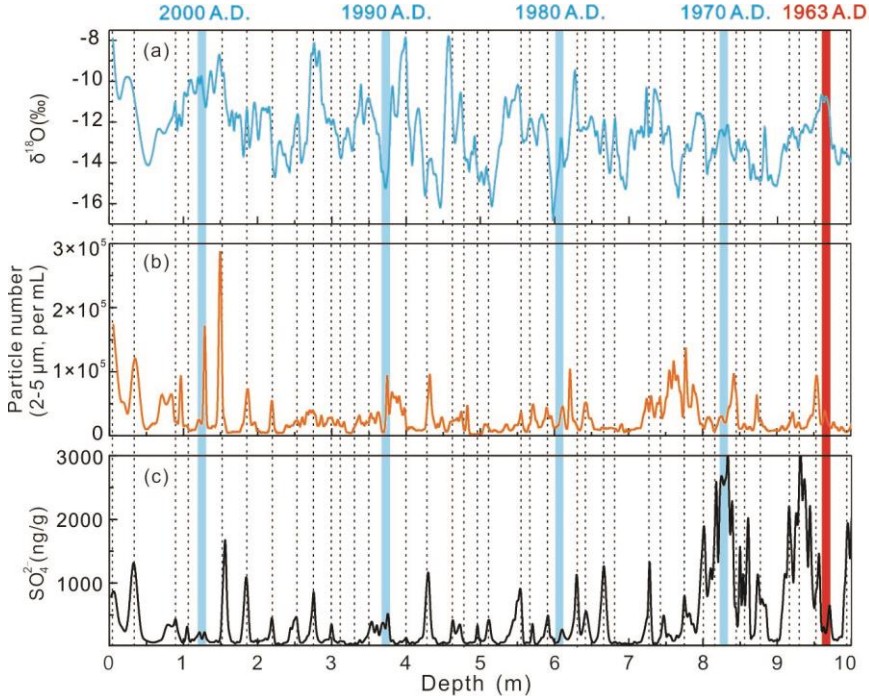

**Figure 2: Dating results of the upper 10 m Tanggula ice core by layer counting. (a) δ¹⁸O of ice. (b) Concentration of insoluble particulates with grain size between 2 and 5 μm, (c) Sulfate ion concentration. The red vertical bar indicates the β activation maximum at 1963 A.D. in the TP.**

The age of the upper 27.13 m Tanggula ice core was determined by counting the annual variations of the physical and chemical parameters with seasonal variation characteristics in the ice. Based on the previous ice core chronology works in the Tanggula glacier (e.g., Zheng, et al., 2010; Joswiak, et al., 2010), variations of the δ¹⁸O of ice, the concentrations of the sulfate ion and insoluble particulate in ice were compared together to determine each annual layer of the upper 27.13 m ice core (see Fig. 2). At the same time, absolute age control was done to the dating of the Tanggula ice core. The maximum radioactive β activation point in the TP which occurred around 1963 A.D. (Thompson, et al., 2006) was used to constrain the layer counting results of the ice core (Fig. 2, red bar). It showed that the layer counting results of the upper ice core agreed well with the absolute dating by the radioactive β activation. Finally, the age span of the upper 27.13 m ice core was determined, which ranged from 1859 to 2004 A.D.. We conclude that the accuracy of the dating results is within 2 years.

The same annual layer counting method was used to date the ice core from 27.13 to 92.14 m, except that only the δ¹⁸O of ice and the insoluble particulate concentration were compared together to determine each annual layer. By this, the age span of the ice core from 27.13 to 92.14 m was determined to be from about 1024 to 1858 A.D.. Below 92.14 m, the annual signals in the ice core were difficult to identify due to the time span of each ice sample was greater than 1 year and the layer counting method cannot be used. Therefore, other dating methods are needed for dating the deeper ice samples.



Generally, dating of the deeper ice core could be calculated by the glacier flow model (GFM), or by comparing the physical or chemical parameter in the ice core with other dated variables with close relationship (e.g., Parrenin et al., 2007; Bazin et al., 2013; Jouzel, 2013). For the GFM method, because the Tanggula ice core did not reach bedrock, the exact thickness of the glacier is unknown. This could lead to great difference in the calculation of age-depth relationship. Consequently, further dating method was needed to construct the age-depth relationship for the deeper part of the Tanggula ice core. As according

to previous studies about the ice core air bubbles, variation of the air content in ice core agreed well with that of the total solar irradiation (TSI) in polar (e.g., Raynaud et al., 2007; Eicher et al., 2016) and high-altitude alpine glaciers (Li et al., 2011). Thus, we could date the age of the ice core at different depth by comparing the gas content in ice core with the TSI that known the age (same method could be seen in Bazin et al., 2013). In order to verify the applicability of this method on the Tanggula ice core, we firstly compared the ice core air content record over the past 1000 years based on the chronology

results by layer counting with the TSI in the NH (Fig. 3a). It showed that although there were some slight differences in the variation phase between these two variables, their general variation characteristics were in good agreement with each other over the past millennium. This indicates that it is available to establish the age-depth relationship of the deep part of the Tanggula ice core by comparing the variation of the ice core air content with that of the TSI in the NH. Consequently, based on this method, 60 age control points were obtained and the general age-depth relationship of the 190.3 m Tanggula ice core

was established (black box points in Fig. 3b).

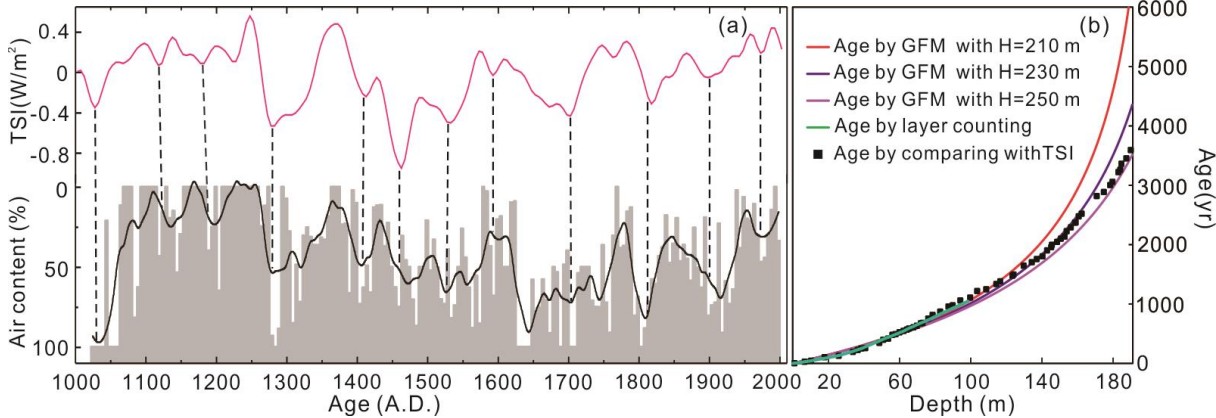

**Figure 3: Dating results of the Tanggula ice core. (a) Comparison of air content in the Tanggula ice core with the total solar irradiation (TSI) in the NH over the past 1000 years. The black line is the 5-point moving average of air content in the Tanggula ice core. (b) Age-depth relationships established by layer counting, GFM and comparison with TSI.**

We compared the age-depth relationship results of the Tanggula ice core established by layer counting, the GFM (detailed calculation process was given by Bolzan in 1985) and the comparison with TSI (Fig. 3b). It shows that they are in good agreement with each other within the upper 90 m ice core, which confirmed that the dating results obtained by layer counting was credible. On the other hand, these five age-depth relationship curves begin to diverge with depth increase under 90 m. The age calculated by the GFM varied greatly with glacier thickness ($H$). Fortunately, the age-depth relationship determined

by comparing with the TSI was in good agreement with that determined by the GFM when the glacier thickness ($H$) was





assumed as 250 m. Their consistency indicated that the general age-depth relationship determined by comparing with the TSI was credible. Although it might have some difference between the exact ages and ages calculated by the age-depth relationship established by comparing with the TSI, it could be used to determine the general age of the deep Tanggula ice core. Thus, combined with the dating result of the upper 92.14 m ice core by layer counting and the dating result of the ice

under 92.14 m by the reconstructed age-depth relationship established by comparing with the TSI, the chronology of the 190.3 m Tanggula ice core were determined. The result showed that the age span of the 190.3 m ice core was ~3600 years from 1610 B.C. to 2004 A.D. (corresponding to the late Holocene). The chronology of the ice core were used as the chronology of the ice core air bubbles according to discussion about the gas-ice age relationship in the Tanggula glacier above.

**3.2 Implication of the variation of δ18Obub in the Tanggula ice core**

In this study, detections of the modern natural atmosphere collected from six locations inside and outside of the TP showed that there was no difference in the $\delta^{18}O_{atm}$ of the modern natural atmosphere (Table.1). All of the detected $\delta^{18}O_{atm}$ values were around -0.37‰. This is consistent with the previous studies (e.g., Raynaud et al., 2007; Luz and Eugeni, 2011; Hou et al., 2013) which showed that there was no difference in the $\delta^{18}O_{atm}$ around the world. The $\delta^{18}O_{atm}$ of the modern natural

atmosphere worldwide now is about -0.05‰ (e.g., Bazin et al., 2013; Extier et al., 2018). The 0.32‰ difference between the $\delta^{18}O_{atm}$ detected in this study and those in polar glaciers is due to the different $\delta^{18}O_{atm}$ value of reference gases assigned during the stable isotope detections in different labs. As there is no difference in the $\delta^{18}O_{atm}$ of the modern natural atmosphere all around the world, according to this we adjusted the $\delta^{18}O_{atm}$ value of the reference gas used in this study to those in the polar glaciers for convenience of further discussion about the $\delta^{18}O_{bub}$ results. The final normalized detection

results of the $\delta^{18}O_{bub}$ in the Tanggula ice core were shown in Fig. 4.

**Table 1. δ¹⁸Oₐₜₘ of modern natural atmosphere inside and outside of the TP**

| Sampling Site | Latitude | Longitude | Altitude (m a.s.l.) | δ¹⁸Oₐₜₘ (‰) |
|---|---|---|---|---|
| Beijing City | 116.39 °E | 39.99 °N | 52 | -0.4‰ |
| Lanzhou City | 103.85 °E | 36.06 °N | 1590 | -0.39‰ |
| Lhasa City | 91.02 °E | 29.64 °N | 3660 | -0.36‰ |
| North slope of Mt. Everest | 86.56 °E | 28.21 °N | 4276 | -0.39‰ |
| Geladandong Glacier | 91.07 °E | 33.63 °N | 5185 | -0.38‰ |
| Qiangyong Glacier | 90.22 °E | 28.87 °N | 5325 | -0.34‰ |

Figure 4 shows that the $\delta^{18}O_{bub}$ in the Tanggula ice core (red line) had undergone significant changes since the late Holocene. It fluctuated significantly around the $\delta^{18}O_{atm}$ of modern natural atmosphere (about -0.05‰, black dotted line). The fluctuation range of the $\delta^{18}O_{bub}$ in the Tanggula ice core (exceeded 6‰) was much larger than that of the variation range of



$\delta^{18}O_{atm}$ in the natural atmosphere caused by variations in the productivities of global continental and oceanic biosphere or the sea level (about 0.7‰, the yellow shaded area) even since the Holocene (e.g., Severinghaus et al., 2009; Bazin et al., 2013; Extier et al., 2018). This indicates that variation of the $\delta^{18}O_{bub}$ in the Tanggula ice core should be affected by other factors rather than the global climate changes.

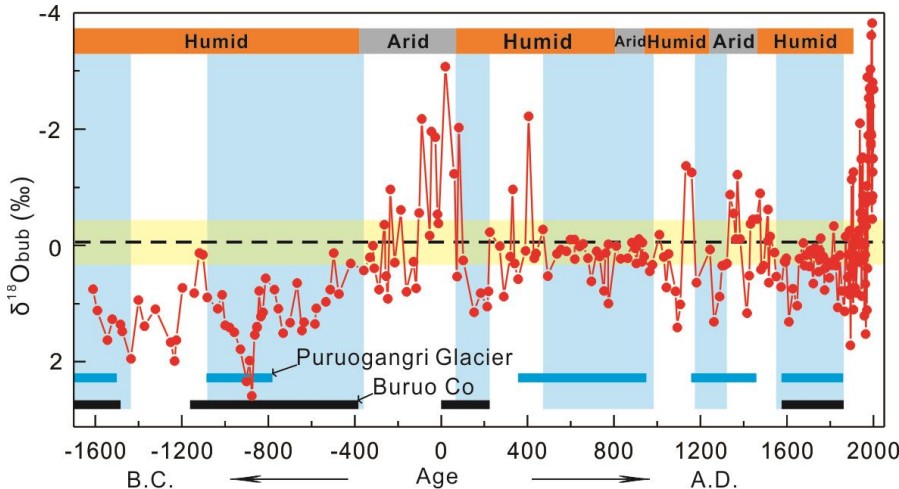

**Figure 4: Variation of the $\delta^{18}O_{bub}$ in the Tanggula ice core since the late Holocene. The Orange and gray horizontal bars at the top indicate the intensity of the Indian monsoon (Chauhan et al., 2010). The yellow shaded area indicates the fluctuation range of $\delta^{18}O_{atm}$ in the natural atmosphere since the Holocene. The blue and black horizontal bars at the bottom indicate the cold events recorded by the Puruogangri ice core (Thompson et al., 2006) and Buruo Co lake sediment (Xu et al., 2019).**

As previous studies had shown that except for the global climate changes, variation of the $\delta^{18}O_{bub}$ in ice core could be affected by gravitational or thermal fractionation of the oxygen isotope in firn layer (Huber et al., 2006). For example, in dry recrystallization-ice formation zone, the firn layer on the glacier surface is very thick and reaches tens of meters. The gravitational fractionation of the oxygen isotope in firn could yield a positive $\delta^{18}O_{bub}$ of the gaseous oxygen in ice. In addition, in the percolation-recrystallization zone, the oxygen isotope of gases in firn layer or ice could exchange with that of

the melt water through a series of physical and chemical reactions caused by strong ultraviolet rays before or after the enclosure of gases into ice (Lunak and Sediak, 1992; Luz and Eugeni, 2011; Mader et al., 2017). This exchange process would lead to the negative variation of the $\delta^{18}O_{bub}$ in the ice due to that the $\delta^{18}O$ of meltwater is more negative than the $\delta^{18}O_{atm}$ of the atmosphere. Thus, considering the significant variation of the $\delta^{18}O_{bub}$ in the Tanggula ice core, we concluded that its variation should be closely related to the variation of the ice formation process on the glacier surface under different

regional climate conditions.

In Fig. 4, we could see that there were six positive and five negative $\delta^{18}O_{bub}$ stages in the Tanggula ice core. When the $\delta^{18}O_{bub}$ were positive, the temperature in the central TP were also very cold (light blue vertical bars, Thompson et al., 2006; Xu et al., 2019). Meanwhile, according to Chauhan et al. (2010) the Indian monsoon were at strong stages and could bring more precipitation to the TP. Therefore, under the low-temperature conditions, the increase of precipitation helped the





development of thick snow layer on the surface of the Tanggula glacier. The positive $\delta^{18}O_{bub}$ values should be due to the gravitational fractionation of the oxygen isotope in the firn. The longer the negative temperature environment persists, the thicker the firn and the more positive the $\delta^{18}O_{bub}$ in the Tanggula ice core. Based on the observed gravitational fractionation rate of the gaseous oxygen isotope in glacial firn by Huber et al. (2006), the $\delta^{18}O_{bub}$ in the Tanggula ice core and the $\delta^{18}O_{atm}$ of natural atmosphere, it could be calculated that the firn thickness at the ice core drilling site of the Tanggula glacier had

exceed 70 m during 1620-450 B.C.. After that, there were no such thick firn anymore according to the variation of the $\delta^{18}O_{bub}$.

On the other hand, when the $\delta^{18}O_{bub}$ in the Tanggula ice core were negative, the regional temperature were relatively warm. The negative variation of the $\delta^{18}O_{bub}$ should be due to the exchanges of oxygen isotopes between gases and meltwater through a series of physical and chemical processes via strong ultraviolet radiation on the glacier surface at high altitude. As

the $\delta^{18}O$ of meltwater is more negative than the atmospheric $\delta^{18}O_{atm}$, the isotope exchanges resulted in negative $\delta^{18}O_{bub}$ values in the ice core. The longer the positive temperature environment persists every year, the greater the melting and infiltration of the firn and the more negative the $\delta^{18}O_{bub}$ in the Tanggula ice core. For instance, over the last 100 years, the $\delta^{18}O_{bub}$ in the Tanggula ice core significantly decreased. It was consistent with the recent dramatic global warming and the increased melting of glaciers in the central TP (Yao et al, 2012). Although the exchange of oxygen isotopes between gases

and meltwater in the firn occurred several times since the late Holocene, this process have not reached equilibrium yet. The most positive $\delta^{18}O$ value of the ice (-9.89‰) is still much lower than the most negative $\delta^{18}O_{bub}$ value in the Tanggula ice core. This indicates that even during the warmest period since the late Holocene, the persist time of the positive temperature in each year was still not long enough for the complete exchange of oxygen isotopes between the gases and the meltwater in the firn.

**3.3 Late Holocene glacier variations in the central TP**

As according to the discussions above, variation of the $\delta^{18}O_{bub}$ in the Tanggula ice core was dominated by the accumulation or melting (deficit) of the glacier. Then, it should be used to reveal the regional glacier variations, especially the obvious accumulation or deficit events. When the $\delta^{18}O_{bub}$ in the ice core was positive, it indicated that the Tanggula glacier accumulated. When the $\delta^{18}O_{bub}$ in the ice core was negative, it indicated that the Tanggula glacier suffered deficit. In Fig. 5,

we could see that since the late Holocene, there were four obvious accumulation and three deficit periods of the glaciers in the central TP indicated by the variations of the $\delta^{18}O_{bub}$ in the Tanggula ice core (Fig. 5, d). The strongest accumulated period was from 1610-450 B.C., which is consistent with the significant glacier advance during this period at the Mt. Tanggula (Deng and Zhang, 1992; Li and Li, 1992). It could also be divided into two stages, which is 1610-1200 B.C. and 1090-450 B.C., respectively. The third glacier accumulation period was from 200-300 A.D., which was relatively short. The

most recent ice accumulation period was 1230-1900 A.D., during which the accumulation strength was relatively weak and even deficit stages presented. The most significant deficit period of the glaciers in the central TP was the last 100 years, and the other two significant deficit periods were 300 B.C.-200 A.D. and 300 A.D.-1230 A.D..



To further understand the relationship of late Holocene glacier variations in the central TP to regional and hemispheric climate changes, we compared them with those of the Westly dominated Muztagh Ata glacier (Fig.5, c, Liu et al., 2014), the

monsoonal temperate glaciers at the Greater Himalaya in central Garwhal (Fig.5, f, Murari et al., 2014) and at the Bhutanese Himalaya (Fig.5, g, Xu et al., 2020) and the glaciers in the west-central Europe (Fig.5, b, Holzhauser, et al., 2005), the floating ice events indicated by the sediment record of hematite-stained grains in the North Atlantic (Fig.5, a, negatively related to the North Atlantic Oscillation, Bond et al., 2001) and also temperature reconstructions in the eastern TP (Fig.5, e, Hou et al, 2016), the Europe (Fig.5, pink and blue bars, Lamb 1985; Chauhan et al., 2010)  and the NH (Fig.5, h, Kobashi et

al., 2013).

The results showed that the significant glacier accumulation periods in the central TP during 1610 B.C.-1200 B.C. and 1090-450 B.C. were consistent with the significant increases of floating ice events in the North Atlantic. It corresponds to the Neoglaciation in the TP (e.g., Zheng et al., 1997; Owen et al., 2005; Yi et al., 2008). During this period, the monsoonal temperate glaciers in the Himalaya advanced and temperature in the Europe and the eastern TP was also very low. Glaciers

in the Westly dominated Muztagh Ata and the west-central Europe did not advance at the first accumulation period, but advanced during 1090-450 B.C. which was consistent with the temperature decrease in the NH.

During the relatively short accumulation period in the central TP from 200 A.D. to 300 A.D., glaciers in the Himalaya, the Westly dominated Muztagh Ata and the west-central Europe also advanced. These glacier advance events were consistent with the temperature decrease in the eastern TP and the NH. This cold event was also consistent with the temperature

decrease in the eastern TP (Liu et al., 2009) and the glacier retreat in the southern TP (Zhang et al., 2017). During the Little Ice Age (LIA) (Mann, 1999; Jones and Mann, 2004), the glacier accumulation in the central TP was consistent with the increase of floating ice events in the North Atlantic. During this period, glaciers in the Himalaya, the Westly dominated Muztagh Ata and the west-central Europe advanced, and temperature in the eastern TP decreased. The short deficit events occurred during this general cold period was in good agreement with the intervallic temperature increase in the NH.





**Figure 5: Comparisons of glacier variations in the central TP (d) with variations of the Westly dominated Muztagh Ata glacier (c, blue and brown bars represent advance and retreat, respectively.), advances of monsoonal temperate glaciers in the Greater Himalaya in central Garwhal (f) and the Bhutanese Himalaya (g), glacier variations in the west-central Europe (b), variations of hematite-stained grains in the North Atlantic (a, reversed vertical axis.) and temperature reconstructions in the eastern TP (e), the Europe and the NH (h). The pink and blue bars at the top indicate climatic events in the Europe: HSACP = High Sub Atlantic Cold Period, RWP = Roman Warm Period, CDA = Cold Dark Age, MWP = Medieval Warm Period, LIA = Little Ice Age, and RWA = Recent Warming Age.**

From 300 B.C. to 200 A.D. when the Europe was at the Roman Warm Period (RWP), the glaciers in the central TP, the Westly dominated Muztagh Ata, the Himalaya and the west-central Europe consistently retreated. The intensity of the North Atlantic Oscillation (NAO) also increased. Temperature in the eastern TP and the NH were high. These indicated a synchronous warm climate in the NH. From 300 A.D. to 1230 A.D., the glacier deficit in the central TP was not in good agreement with variations of the glaciers in the Muztagh Ata, the Himalaya and the west-central Europe. It was more consistent with the variations of the NAO and the temperature in the eastern TP. Previous studies in the western (Yao et al., 1997), eastern (Liu et al., 2009) and southern TP (Zhang et al., 2017) also showed that the TP was in a warm period from



300 A.D. to 500 A.D.. Meanwhile, from 1100 to 1200 A.D. when the NH was at the Medieval Warm Period (MWP) (Mann, 1999; Jones and Mann, 2004), many regions of the TP were at warm stage too (e.g., Yang et al., 2003; Zhang et al., 2017). Over the last 100 years, the significant deficit in the central TP indicated a dramatic increase of regional glacial melting. The amplitude of the glacial melting were the strongest over the past 3600 years. This is consistent with the rapid melting of glaciers in the TP (especially in the southern TP) (Yao et al., 2012), the glacier retreats in the Westly dominated Muztagh

Ata and the west-central Europe corresponding to the recent warming age (RWA) in the Europe and the NH. The NAO increase trend was also rapid during this period.

Comparison results showed that variations of the glaciers in the central TP since the late Holocene were more consistent with those of the floating ice events in the North Atlantic. This indicated a close relationship of the late Holocene glacier variations in the central TP to the variations of the NAO. This result is consistent with those previous climatic and

environmental studies in the TP, which showed that there was closely relationship between the climate changes in the TP and the NAO (e.g., Chen et al., 2001; Zhu et al., 2015; Xu et al., 2019).

## 4 Conclusions

In this study, a 190.3 m ice core drilled from the Tanggula glacier in the central TP was investigated. Dating results show that the age span of this ice core was ~3600 years from 1610 B.C. to 2004 A.D. which could also be used as the chronology

of air bubbles enclosed in ice when discussing its millennial variations. The consistency of the variations of air content in the Tanggula ice core with total solar intensity showed that it was an available method for dating the deep parts of the ice cores in the TP. Based on the analysis of the ice formation process on the Tanggula glacier surface and the influencing factors of the $\delta^{18}O_{bub}$ in glacial ice, we concluded that there was a good correlation between the variation of $\delta^{18}O_{bub}$ in the Tanggula ice core and the accumulation or deficit of the glacier. During the warm period, when there were melting on the glacier, the

$\delta^{18}O_{bub}$ was more negative than the natural atmospheric $\delta^{18}O_{atm}$. During the cold period, when the firn accumulated on the glacier, the $\delta^{18}O_{bub}$ was more positive. Variations of the $\delta^{18}O_{bub}$ in the Tanggula ice core indicated that over about the past 3600 years, there were four accumulation and three deficit periods of the glaciers in the central TP. The strongest accumulation period was from 1610-450 B.C. which could be divided into two separated stages. The other two glacier accumulation periods were 200-300 A.D. and 1230-1900 A.D.. The most significant deficit period of the glaciers in the

central TP was the last 100 years, and the other two deficit periods were 300 B.C.-200 A.D. and 300 A.D.-1230 A.D.. Glacier variations in the central TP since the late Holocene were closely related to the climate changes in the high latitudes of the NH, such as the NAO.

## Acknowledgments

This work was supported by the National Key R&D Program of China (2018YFB1307504), the National Natural Science

Foundation of China (41201058) and the Strategic Priority Research Program of Chinese Academy of Sciences



(XDA20070102). We thank all of the field workers for the hard work of ice core collection on the glacier. We also thank Dr. Chenglong Zhang of Research Center for Eco-Environmental Sciences, Chinese Academy of Sciences for the help of ice core pretreatments in the cold room. Many thanks are also given to Dr. Dongmei Qu and Dr. Shaopeng Gao of the Institute of Tibetan Plateau Research Chinese Academy of Sciences for their help in the determination of the ice core air bubbles.

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
