# Peer review of "Late Holocene glacier variations in the central Tibetan Plateau indicated by the $\delta^{18}$ O of ice core enclosed gaseous oxygen"

_The Cryosphere, 2022_

## Referee Comment (RC2)

Review of "Late Holocene glacier variations in the central Tibetan Plateau indicated by the δ18O of ice core enclosed gaseous oxygen." by Jacob J. Li et al, The Cryosphere.

**General:**

The manuscript presents new results of the isotopic composition of molecular oxygen ($\delta^{18}O_{bub}$) enclosed in the alpine glacier Tanggula in the central Tibetan Plateau. They document that there is a good correlation between the these measurements and the accumulation rate or melting rate of the glacier. Deeper firn layers go along positive $\delta^{18}O_{bub}$ values, whereas stronger glacier melting goes along with negative $^{18}O_{bub}$ values. Based on these findings, they discuss variations of the glacier and found four accumulation driven and three melt driven episodes. They also state that their measurements support a link between climate variations on the Tibetan Plateau and the North Atlantic Oscillation.

Major points:

There are major shortcomings that forces me to reject this manuscript for publications in its present form. The introduction is not focussed on the subject discussed on the paper but lists many areas of research done with these measurements at different places not comparable with alpine ice. No discussion is made about potential influences on d18Obub from processes such as firn thickness variations, melt water influences etc. Furthermore, the experimental setup is not well explained and lacks important information such as which absolute horizons have been used for dating. The author state that the dating is very robust to +-2year, but no clear discussion is made that supports this statement. There is a good possibility to measure the air content of the ice when extracting the air from the ice. Has this be done, if not why? This would have been a possibility to compare it with the rough estimate based on images. It is also not clear from the description in section 2.2 what the exact procedure for the measurements of d18Obub is. Was it a online system or has the air been collected on a cold trap? Was the air further cleaned from water vapour, nitrogen, argon, CO2 etc or not? A scheme of the setup would be helpful here. Was only d18O measured or also other species (isotope and elemental ratios)?
Furthermore, nothing is said about how the glacier flow model lines are obtained (Fig. 3). No explanation of the model, the assumptions to run it etc.

Intro:     The introduction is rather general and does summarize for what d18O on molecular oxygen has been and can be used. Yet, limited information is given about measurements on alpine glaciers in general. For instance influences of melt water on firn thickness, chemical reactions initiated through melt water and influences on gas diffusion etc. All these processes might be relevant for the observed variations of the isotopic composition of molecular oxygen enclosed in ice. Therefore, I would suggest to focus in the introduction to items relevant for the manuscript.

Along with d18O generally also d15N is measured. Do the author have these measurements?

Line 40f:  Regarding the Dole effect: important contributions are missing

Line 49ff: I would rewrite this sentence to::
           On the other hand, since the ice formation process in alpine glaciers is more
           sensitive to regional climate conditions and its variations could lead to changes in

the air component in the ice core, the climatic significance of the δ18Obub in alpine glaciers may be different from that in polar glaciers (Luz and Eugeni, 2011). However, few detailed studies have been carried out on this assumption.

Line 64: What about firn thickness and its variation over time. Are there any clues? Do you have measurements of d15N along those of d18Obub? This would be an indication of fractionation processes originating on site.

Line 95f: This means that the air content has not been measured but only roughly estimated based on images?

Fig. 2: Dating seems to be very vague and unclear. More information is required. It is helpful though to have the annual lines aligning the peaks of the three parameters. But there seems to be quite some uncertainty, at least in my view, that surpasses +-2 years stated.

Minor points:

Line 158: ….to large differences …

Fig. 3: Nothing is said about how the glacier flow model lines are obtained. No explanation of the model, the assumptions to run it etc.

Table 1: These measurements are reported against the lab internal reference standard. Please note this explicitly. What kind of standard is it, name it, is it pure ambient air, artificial air or what?

Line 206: you mean +-0.35 promille

Fig. 4: How has the zero line be defined? Generally, one assumes today's ambient air to be zero! The mean d18Obub value of the recent ice is 4 permil depleted!! How come? The offset of your ambient air on your reference is 0.32 permil.

since the Holocene…
you mean during the late Holocene

Line 215f: so what, how deep is the firn layer? Have you calculated the expected gravitational settling effect?

Line 221f: So what, how important is it?

Line 232f: not clear at all

Fig. 5: zu (h) what is the reference to these data?

Line 309: you may rewrite:
This result is consistent with those previous climatic and environmental studies in the TP, which showed that there is a close relationship between climate changes in the TP and the NAO (e.g., Chen et al., 2001; Zhu et al., 2015; Xu et al., 2019).

Line 312: The conclusion section is poor.

---

## Author Comment (AC2)

**Thank you for your attention to the manuscript.**

**General:**

Based on our experiments and analysis, we found a close relationship between the $\delta^{18}O_{bub}$ enclosed in the alpine glacier Tanggula and its variation (strong accumulation or melting) in the central Tibetan Plateau. It showed that the strong accumulation (deep firn) or melting (water involved ice formation) of the glacier could lead to unusual positive or negative values of the $\delta^{18}O_{bub}$. This is the first investigation of long-term series of $\delta^{18}O_{bub}$ enclosed in the Tibetan Plateau glaciers. We think it might could provide an available way to reconstruct the glacier variation and the climate change over the historical time in the third pole Tibetan Plateau.

**Reply to major point comment:**

At the very beginning of the introduction part, we briefly introduced the significance of the ice core air bubbles research for the climate and environment reconstruction. We think this might provide a background of the ice core air bubble research and could make it easier for more readers to understand the detail subject we mentioned next in the manuscript.

We also think these measurement and conclusion results about the variations of component and composition of air bubbles enclosed in glaciers in the polar regions could help for further understanding of what caused the variations of component and composition of air bubbles enclosed in the alpine glaciers.

Discussion about the potential influences on $\delta^{18}O_{bub}$ from processes such as firn thickness variations, melt water influences etc. is firstly introduced in the second paragraph at the introduction part. Then, further discussion was made during the analysis of the implication of the variation of $\delta^{18}O_{bub}$ in the Tanggula ice core.

Dating of the ice core in this manuscript was based on the combination of different approaches. The upper (shallow) part of the ice core was dated by the annual counting method which was widely used for the alpine glaciers in The Tibetan Plateau. Then the dating model for the mountain glaciers which was developed by other researcher Bolzan et al. (1985) was used for the deeper part of the ice core when the annual signal in not clear. Absolute age control of the annual counting result was done by the measurement result of the $\beta$ activation maximum at 1963 A.D. in the TP. Through the comparisons between these two results, we concluded that the uncertainty should be about 2 years. For deeper ice, the credibility of the dating result calculated by the Bolzan ice flow model was verified by the dating result from comparisons of ice core air content to the solar variation. More detailed and accurate description will be given in the revised draft.

The air in the ice was not extracted from the ice. On the other hand, we used an image analysis method during the describing process of the ice core physical properties. The results was used in this manuscript for the roughly dating of the ice core by compared it the solar variation, Which could help to calibrate the model calculated age. The detailed method was described in section 2.2.

The procedure for the measurements of $\delta^{18}O_{bub}$ was described in section 2.2. As the release method of the air in the ice was a mature method developed by our coauthor and was already published, so we did not describe it in this manuscript, but cited. It was an online system. And the water vapor was firstly cleaned in the cold trap after the air released. Furthermore, it was secondly cleaned by a nafion drying tube before the air component was transported into the Mat-253. The $\delta^{18}O_{bub}$ and $\delta^{17}O_{bub}$ were both measured by the Mat-253. No other isotope and elemental ratios were measured during the experiment.

The glacier flow model formula was obtained from the published article which was cited in the manuscript. So detailed method was not mentioned in our manuscript.

**Reply to Intro:**
Yes we agree. In our discussion part we analysis the physical and chemical influence to the variation of the $\delta^{18}O_{bub}$. More introduction of these items relevant for the manuscript will be added to the revised manuscript. The $\delta^{15}N$ wan not measured in this study.

**Reply to Line 40f:**
More contribution factors like the ecosystem variation to the Dole effect will be added in the revised manuscript.

**Reply to Line 49ff:**
Yes, we agree.

**Reply to Line 64:**
As the $\delta^{15}N$ was not measured for this study, the exact firn thickness was not discussed in the manuscript.

**Reply to Line 95f:**
we used an image analysis method during the describing process of the ice core physical properties. It is used in our manuscript for the roughly dating of the ice core by compared it the solar variation. The detailed method was described in section 2.2.

**Reply to Fig.2:**
Yes, we agree that uncertainty should be existed. The absolute age control of the annual counting result was done by the measurement result of the β activation maximum at 1963 A.D. in the TP. Through the result comparison, we concluded that the uncertainty should be about 2 years. For deeper ice, the dating result calculated by the Bolzan ice flow model was verified by the roughly dating result from comparison of ice core air content to the solar variation. It is possible that sometimes there was age errors for some part of the ice core, but it does not affect the result discussion in decadal or centurial scale.

**Minor points:**
**Reply to Line 158:**
Yes, we agree.

**Reply to Fig.3:**

The Bolzan glacier flow model formula was obtained from the published article which was cited in the manuscript. So detailed method was not mentioned in our manuscript.

**Reply to Table 1:**

The lab internal reference standard gas in this manuscript is the compressed ambient air which was mentioned in the section 2.2. More information will be added in the revised manuscript.

**Reply to Line 206:**

Yes, we agree.

**Reply to Fig.4:**

In our experiment we assume today's ambient air to be zero. Yes. We think the unusual variation of the $\delta^{18}O_{bub}$ in the Tanggula ice core should be affected by other factors rather than the global climate changes. Based on our analysis we concluded that the oxygen isotope of gases in firn layer or ice could exchange with that of the melt water through a series of physical and chemical reactions caused by strong ultraviolet rays before or after the enclosure of gases into ice.
Yes, we mean during the late Holocene.

**Reply to Line 215f:**

As the $\delta^{15}N$ was not measured for this study, the exact firn thickness was not discussed in the manuscript. However isotopic gravity fractionation in firn has been demonstrated in other studies. It was cited in the introduction part.

**Reply to Line 221f:**

Yes, we think it is important. This also showed the different climatic and environmental implications of ice core gas isotopes of glaciers in the TP compared to that in the Polar regions.

**Reply to Line 232f:**

We will rewrite it in the revised manuscript to make it for better understanding.

**Reply to Fig.5:**

The reference to these data was cited from Line 264 to Line 270.

**Reply to Line 309:**

Yes, we agree.

**Reply to Line 312:**

More conclusion detail will be added in this part in the revised manuscript.

---

## Author Comment (AC3)

**Thank you for your attention to the manuscript.**

First, the analytical method is not fully described, and some of the results are very hard to understand if all the analyses are accurate.

**Reply:**
**The measurement methods of these data used in the manuscript were described in section 2.2. According to this comment we will describe more analytical method in the revised manuscript for better understanding.**

Second, the paper deduces correlations between various climate records, but does not validate these correlations statistically.

**Reply:**
**Through our research, we found the relationship between the $\delta^{18}O_{bub}$ and the variation of the alpine glacier Tanggula. This feature might help us to reconstruct the glacier variation trend over the past, although the statistically analysis were not made. We think it should be of great significance to carry out digital analysis between sequences based on more accurate chronological data in the further work.**

Third, some basic physics is invoked but not described. For example, there is no explanation for how water and O2 can exchange isotopes fast enough to influence the isotopic composition of O2 in trapped gases.

**Reply:**
**The main purpose of this manuscript is to present the relationship between the $\delta^{18}O_{bub}$ and the variation of the alpine glacier Tanggula. This conclusion was confirmed after comprehensive analysis and comparison with regional glacier changes, although we do not understand the exact exchange progress right now.**

Specific concerns include:
Total air content was determined by an indirect qualitative method and was apparently not checked against robust observations. (Section 2.2)

**Reply:**
**The air content in the Tanggula ice core was determined by describing the characteristics of how much bubbles in a restricted area in the computer. This method maybe not that accurate, but it is the real character of how much bubbles in the ice. We think the result should be used to deduce the variation trend of the air content in the ice core.**

In Figure 2, the data does not constrain annual layer thicknesses well. Bomb radioisotopes are invoked but the data are not included.

**Reply:**
**The radionuclide β activation in the ice core was detected using a low-concentration α-β activation counting instrument at the SKLCS for absolute age control of the ice. It shows the maximum at the 9.6m of the ice core. So we put it in Fig.2. The detail message will be added in the revised manuscript.**

The analytical method for measuring and standardizing d18O of O2 was not fully described.

**Reply:**
**More information about the measuring and standardizing $d^{18}O$ of O2 will be added in the revised manuscript.**

Figure 3: The relation between TSI and air content is not validated by a simple x-y plot showing the relationship between the 2 properties, or other approaches. The high value for air content comes around 1640, but there is no TSI maximum at this time.

**Reply:**
**The comparison between the TSI and air content was used for age control of the age-depth relationship calculated by the Bolzan flow model. As the air content was not the exact value. There could be some deviations between these two variables. But the overall trends and characteristics should be consistent.**

Table 1: the authors do not explain how they measured d15N, which is needed to calculate d18Oatm. I could not find information about the reference gas.

**Reply:**
**The $\delta^{15}N$ wan not measured in this study. The reference gas was the compressed ambient air which was mentioned in the section 2.2.**

Figure 4: there is no statistical documentation for a relationship between climate and d18Obub. Also in Fig. 4: d18Obub reaches +2 per mil, which would require a firn column thickness of about 200 meters thick at certain times. This seems unlikely to say the least.

**Reply:**
**Yes, the $\delta^{18}O_{bub}$ was relatively high compared to the that in the atmosphere. As the δ15N wan not measured in this study, we could not calculated how deep is the firn layer.**

Fig. 5. There is no statistical evidence showing coherency between Tangguula and other records.

**Reply:**
**This figure showed that the variation of the glacier in the central Tibetan Plateau was not quite the same with those in the other region. But it indicated a close relationship of the late Holocene glacier variations in the central TP to the variations of the NAO.**

Lines 220-225: There is no evidence that water and O2 exchange isotopes fast enough to impact the isotopic composition of O2 in ice cores. At least the authors do not make a case that extensive exchange is plausible.

**Reply:**
**In this manuscript discussion part, we analysis the physical and chemical influence to the variation of the $\delta^{18}O_{bub}$. But we did no further research. In the next work we will focus on the progress and mechanism of the isotopic exchange during or after the storage of the air into the ice.**